

# 1  Tropospheric column ozone response to ENSO in GEOS-5
# 2  assimilation of OMI and MLS ozone data

**Mark A. Olsen[1,2], Krzysztof Wargan[3,4], and Steven Pawson[3]**
[1] Atmospheric Chemistry and Dynamics Laboratory, Code 614, NASA Goddard Space Flight
Center, Greenbelt, MD
[2] Goddard Earth Science, Technology and Research Center, Morgan State University, Baltimore,
MD.
[3] Global Modeling and Assimilation Office, Code 610.1, NASA Goddard Space Flight Center,
Greenbelt, MD
[4] Science Systems and Applications Inc., Lanham, MD
Correspondence to: M. A. Olsen (mark.olsen@nasa.gov)

## 14  Abstract

We use GEOS-5 analyses of Ozone Monitoring Instrument (OMI) and Microwave Limb Sounder
(MLS) ozone observations to investigate the magnitude and spatial distribution of the El Niño
Southern Oscillation (ENSO) influence on tropospheric column ozone (TCO) into the middle
latitudes.  This study provides the first explicit spatially resolved characterization of the ENSO
influence and demonstrates coherent patterns and teleconnections impacting the TCO in the
extratropics.  The response is evaluated and characterized by both the variance explained and
sensitivity of TCO to the Niño 3.4 index.  The tropospheric response in the tropics agrees well
with previous studies and verifies the analyses.  However, we show a newly identified two-lobed
response symmetric about the Equator in the western Pacific/Indonesian region consistent with
the large-scale vertical transport.  We also find that the large-scale transport in the tropics
dominates the response compared to the small-scale convective transport.  The ozone response is
weaker in the middle latitudes, but significant explained variance of the TCO is found over
several small regions, including the central United States.  However, the sensitivity of TCO to the



Niño 3.4 index is statistically significant over a large area of the middle latitudes. The sensitivity
maxima and minima coincide with anomalous anti-cyclonic and cyclonic circulations where the
associated vertical transport is consistent with the sign of the sensitivity. Also, ENSO related
changes to the mean tropopause height can contribute significantly to the midlatitude response.
Comparisons to a 22-year chemical transport model simulation demonstrate that these results
from the nine-year assimilation are representative of the longer-term. This investigation brings
insight to several seemingly disparate prior studies of the El Niño influence on tropospheric
ozone in the middle latitudes.

## 37  1   Introduction

The contributions by natural phenomena to tropospheric ozone variability must be identified and
quantified for robust assessments of the present and future anthropogenic influence. Here, we
investigate the signal of the El Niño Southern Oscillation (ENSO) in extratropical tropospheric
ozone in a global assimilation system. This study provides the first explicit spatially resolved
characterization of the ENSO influence, and reveals coherent patterns and mechanisms of the
influence in the extratropics.
ENSO is well known to impact the magnitude of tropospheric ozone in the tropical Pacific. El
Niño (La Niña) conditions are characterized by anomalous increases (decreases) in SSTs in the
central and eastern Pacific. Opposite anomalies tend to occur in the western Pacific. In general,
changes to convection and circulation patterns under El Niño conditions lead to reduced tropical
tropospheric ozone in the central and eastern Pacific and enhanced ozone over the western Pacific
and Indian Oceans. The response is highly linear in the tropics, so La Niña conditions produce an
antisymmetric response (DeWeaver and Nigam, 2002). This influence on tropical tropospheric
ozone has been observed in satellite data (e.g., Chandra et al., 1998; Thompson et al., 2001;
Ziemke et al., 2010; Ziemke et al., 2015) and ground-based measurements (e.g., Fujiwara et al.,
1999; Lee et al., 2010). Both chemical transport models (CTMs) driven by analyzed meteorology
and free-running models have simulated this impact of ENSO on the tropical ozone (e.g., Sudo
and Takahashi, 2001; Zeng and Pyle, 2005; Doherty et al., 2006; Oman et al., 2011).



The tropospheric ozone response to ENSO in the extratropics has not been as extensively studied
and some results from prior studies appear to be contradictory.  ENSO events have been shown to
alter the extratropical circulation by modifying planetary wave driving, the North Pacific low, and
the location and strength of the extratropical jets (e.g., Angell and Korshover, 1984; Langford,
1999; Trenberth et al., 2002; García-Herrera et al., 2006).  Thus, it is reasonable to expect ENSO
to have a dynamical impact on extratropical tropospheric ozone distribution and variability.
Oman et al. (2013) examined the ozone sensitivity to ENSO with Microwave Limb Sounder
(MLS) and Tropospheric Emission Spectrometer (TES) observations in addition to a chemical-
climate model simulation.  Although limited by just over five years of TES data, they show
statistically significant sensitivity in the lower midlatitude troposphere over two broad meridional
bands centered on the Pacific and Indian Oceans.  Balashov et al. (2014) and Thompson et al.
(2014) find a correlation between ENSO and tropospheric ozone around South Africa using air
quality monitoring station and ozonesonde data.  Langford et al. (1998) and Langford (1999)
show ozone enhancements in the free troposphere correlated with ENSO (with a several month
lag) in lidar data from Boulder, CO.  Langford (1999) attributes this to the secondary circulation
associated with an eastward shifted Pacific subtropical jet exit region under El Niño conditions.
The transverse circulation of ozone-rich air from the stratosphere across the jet is then transported
poleward.  Lin et al. (2015) conclude that more frequent springtime stratospheric intrusions
following La Niña winters contribute to increased ozone at the surface and free troposphere in the
western United States.
In contrast, other observational and modeling studies have not found a significant relationship
between ENSO and extratropical tropospheric ozone, suggesting that any such influence is weak
or occurs only on a regional scale. For example, Vigouroux et al. (2015) use a stepwise multiple
regression model including an ENSO proxy to examine ground-based Fourier transform infrared
(FTIR) measurements from eight subtropical and extratropical stations of the Network for the
Detection of Atmospheric Composition Change (NDACC).  They did not find a significant
ENSO impact on the tropospheric ozone column at any of the eight sites.  Hess et al. (2015) also
did not find a relation between ENSO and tropospheric ozone over extratropical regions in a four-
member ensemble model simulation.  They suggest that ENSO may occasionally induce ozone
anomalies but the correlation is weak.



Determining the spatial extent of ENSO influence on tropospheric ozone from observations is difficult due to the sparse observation networks of sondes, FTIR, etc. The direct retrieval of tropospheric ozone from satellite observations is limited by coarse vertical resolution in the troposphere for nadir-viewing instruments and pressure broadening in the lower troposphere for limb-type instruments. Nevertheless, sonde and surface data combined with satellite observations have been used to derive a coarse global climatology of tropospheric ozone (Logan, 1999). Tropospheric ozone fields have also been derived from subtracting measured stratospheric column ozone from total column ozone (e.g., Fishman et al., 1990; Ziemke et al., 1998; Fishman et al., 2003; Schoeberl et al., 2007). These residual methods are more robust at lower latitudes and have been used to show a large impact by ENSO on tropospheric ozone in the tropics (e.g., Chandra et al., 1998; Ziemke et al., 1998; Thompson and Hudson, 1999; Ziemke and Chandra, 2003; Fishman et al., 2005).

The goal of this paper is to use NASA's Goddard Earth Observing System Version 5 (GEOS-5) analyses of satellite measured ozone to investigate the spatial distribution, magnitude, and attribution of the tropospheric ozone response to ENSO. Assimilation provides the advantages of global, gridded fields constrained by observations. Ziemke et al. (2014) show that the ozone assimilation offers more robust tropospheric ozone fields for science applications in the lower and middle latitudes than residual methods. In the present study, the response in the tropics is evaluated and discussed alongside the midlatitude response. The relatively well-established tropical response is primarily included here for verification of the analyses, although several new findings are discussed. The comprehensive examination of the midlatitudes made possible by the ozone assimilation is novel to this study. In the midlatitudes, the tropospheric column ozone (TCO) is found to have a statistically significant response to ENSO in some regions. This response can be explained by changes to circulation, convection, and tropopause height. These results will benefit both process-oriented evaluations of the regional ozone response in simulations and assessments of the anthropogenic impact on tropospheric ozone, including prediction of future tropospheric ozone and trends.

The following section discusses the data, assimilation system, and methods used in this study. The results are then presented in Section 3. A comparison of results to a CTM simulation is included to show that the nine-year time period of the EOS Aura observations is largely



representative of longer periods.  Additional discussion of the results is found in Section 4 before
concluding with a brief summary.

**2    Data, assimilation system, and methods**
The ozone analyses used in this study were produced using a version of NASA's GEOS-5 data
assimilation system (DAS), ingesting data from the Ozone Monitoring Instrument (OMI) and
MLS on the Earth Observing System Aura satellite (EOS Aura), as described in Wargan et al.
(2015). A brief description of the ozone data and assimilation system is provided in the following
subsection. Subsequent subsections provide information on ancillary data sets used and the linear
regression analysis used in this study.
**2.1    Ozone data and GEOS-5 Data Assimilation System**
The OMI and MLS instruments are both onboard the polar orbiting EOS Aura satellite launched
on July 15, 2004.  OMI is a nadir-viewing instrument that retrieves near-total column ozone
across a 60-scene swath perpendicular to the orbit (Levelt et al., 2006).  The footprint, or spatial
resolution, of the nadir scene is 13 km along the orbital path by 24 km across the track.   The
cross-track scene width increases with distance from nadir to about 180 km at the end rows.  OMI
collection 3, version 8.5 retrieval algorithm data are used in the analyses considered here.   The
MLS instrument scans the atmospheric limb to retrieve the ozone vertical profile from microwave
emissions. Version 3.3 data on the 38 layers between 261 hPa and 0.02 hPa were used in the
present analyses after screening based upon established guidelines (Livesey et al., 2011).
The GEOS-5.7.2 version of the data assimilation system is used to produce the ozone analyses.
This is a modified version from the system used in the Modern-Era Retrospective analysis for
Research and Applications (MERRA) (Rienecker et al., 2011).  For the analyses used here, the
system uses a 2.5°×2.0° longitude-latitude grid with 72 layers from the surface to 0.01 hPa.  The
vertical resolution around the tropopause is about 1 km.  Alongside the ozone data, a large
number of in-situ and space-based observations are included in the GEOS-5 analyses (Wargan et
al., 2015).  However, OMI and MLS ozone retrievals are the only data that directly modify the
analysis ozone in this version of the DAS.  Anthropogenic and biomass burning ozone production





sources are not explicitly implemented in these analyses. However, some impact from emissions
and other tropospheric chemistry sources and sinks is included in the analyses to the extent that
each OMI column retrieval is sensitive to tropospheric altitudes (Wargan et al., 2015).
Wargan et al. (2015) and Ziemke et al. (2014) previously evaluated these ozone analyses relative
to sondes and other satellite data. Their assessments show that accounting for measurement and
model errors in the assimilation greatly increases the precision of the tropospheric ozone over
other methods of obtaining gridded TCO fields. Both Wargan et al. (2015) and Ziemke et al.
(2014) show that there is greater disagreement of the tropospheric ozone analyses with sondes at
high latitudes. For this reason, we restrict our discussion in the present study to the tropics and
middle latitudes.

## 2.2    Global Modeling Initiative CTM simulation

We use a Global Modeling Initiative (GMI) CTM (Strahan et al., 2007; Duncan et al., 2008)
simulation to determine if the results from the nine years of ozone analyses are representative of
the longer term. The simulation is driven using MERRA meteorological fields for 1991-2012
and run at the same resolution as the assimilation system. Observation-based, monthly-varying
surface emissions are used through 2010 with repeated 2010 monthly means for the final two
years. Strode et al. (2015) provide more details on this specific simulation, which they refer to as
the "standard hindcast simulation" in their study. Ziemke et al. (2014) show that the TCO from a
similar GMI simulation compares well with sonde observations. In the present study we define,
process, and analyze the CTM TCO fields in the same manner as the assimilation fields.

## 2.3    ENSO index and outgoing longwave radiation data

ENSO is characterized in this study by the monthly mean Niño 3.4 index available from the
NOAA Climate Prediction Center (http://www.cpc.ncep.noaa.gov/data/indices/). The index is
based upon the mean tropical sea surface temperature between 5° N – 5° S and 170° W – 120° W.
This time series is normalized using 1981-2010 as the base time period. Fig. 1 shows the index
time series from 1991-2013, which spans the years of the ozone analyses and GMI simulation. In
this study, we define "strong" El Niño and La Niña events as months with index values greater
than 0.75 and less than -0.75, respectively. The Climate Prediction Center uses threshold values



of 0.5 and -0.5 to characterize El Niño and La Niña, respectively. The values of ±0.75 used here
to characterize "strong" events is about one standard deviation (0.78) of the time series spanning
the assimilation, 2005-2013. La Niña conditions were dominant during the ozone analyses time
period (black line in Fig. 1). Strong El Niño conditions occurred in the boreal fall/winter of
2006/2007 and 2009/2010. Strong La Niña conditions occurred during the boreal fall/winter of
2005/2006, 2007/2008, 2008/2009, 2010/2011, and 2011/2012.
We use outgoing longwave radiation (OLR) data as a proxy for convection to investigate the
contribution from changes in convection associated with ENSO. The monthly, 1° x 1° data is
provided by the NOAA Earth System Research Laboratory (Lee, 2014). Small values of OLR
indicate substantial convection, and vice versa.
**2.4   Methods**
For the present study, we use the nine full years (2005-2013) of ozone analyses that have been
completed. To calculate the TCO, we define the tropopause at each grid point as the lower of the
380 K potential temperature and 3.5 potential vorticity unit (1 PVU = $10^{-6}$ $m^2$ K $kg^{-1}$ $s^{-1}$) surfaces.
The daily TCO fields are smoothed horizontally by averaging each grid point with the eight
adjacent neighboring points. Monthly mean TCO is computed from the daily values. The large
seasonal variability in the TCO is removed at each point by subtracting the respective nine-year
mean for each month.
We use multiple linear regression of the TCO monthly mean time series onto the Niño 3.4 index
and the first four sine and cosine harmonics to evaluate the response of tropospheric ozone to
ENSO. That is, $TCO = \sum_i m_i X_i + \varepsilon$, where the $X_i$ are the index and harmonic time series, $m_i$
are the best fit regression coefficients, and $\varepsilon$ is the residual error. The regression is computed at
every model grid point. The F-test is used to compute the confidence level of the explained
variances (Draper and Smith, 1998). The calculated significance of the ozone sensitivity includes
the impact from any autocorrelation in the residual time series (Tiao et al., 1990). We find that
tests with time-lagged regressions from one to six months were generally no better than for zero-
lag regressions. Therefore, the results presented herein are computed with no lag of the ozone
time series. This is further discussed in section 4.




## 3 Results

In this section, we examine the magnitude, spatial distribution, and mechanisms of the TCO response to ENSO. For reference, the multi-year annual mean TCO is shown in Fig. 2. The non-seasonal variability is indicated by overlaid contours of one standard deviation of the deseasonalized TCO expressed as a percent of the mean TCO. (Ziemke et al. (2014) illustrate the large seasonal variability). The following two subsections present the explained variance and TCO sensitivity to the Niño 3.4 index. Changes to advection and convection contributing to the TCO response are examined in subsections 3.3 and 3.4. Subsection 3.5 evaluates the ENSO-associated changes to the tropopause height and the impact on the TCO response. We conclude this section with a comparison to CTM results in subsection 3.6 for the purpose of evaluating how robust the results from nine years of ozone assimilation are compared to the longer term.

### 3.1 Explained variance

The percent variance of TCO explained by ENSO is shown in Fig. 3. The ENSO influence is greatest in the tropical Pacific where the variance explained has a maximum of about 55%. This well-known tropical response is associated with increased convection and upwelling in the central and eastern Pacific during El Niño that lofts ozone-poor air into the mid- to upper-troposphere. The anomalous warm ocean current that runs southward along the South American coast during El Niño conditions (e.g., Trenberth, 1997) is evident in the tropospheric ozone response. A northeastward tongue of relatively large magnitude also extends towards and across Central America. An isolated significant maximum is also found between 20° N and 30° N in the subtropical Pacific with explained variance of greater than 20%.

In the western Pacific and Indonesian region, ENSO is known to produce an opposite response to the central and eastern Pacific due to increased upward transport during La Niña conditions. Two lobes of significant explained variance of more than 20% are symmetric around the equator in this region. Off the western coast of Australia, the southern lobe has a maximum of about 35%.

The impact by ENSO is less in the subtropics and middle latitudes compared to the tropical Pacific. Still, the variance explained by ENSO is greater than 20% and statistically significant in





several isolated regions.  Of particular note, the variance explained exceeds 25% over South
Africa and 20% over the central United States.  These areas correspond to locations where
previous studies have found an ENSO signature in ground-based data (Balashov et al., 2014;
Thompson et al., 2014; Langford et al., 1998; Langford, 1999).  The variance explained also
exceeds 20% in a small region south of New Zealand.  Other midlatitude areas, such as the
northern Pacific and Atlantic, exceed 10% but are not statistically significant due to the length of
the time series.
**3.2   TCO sensitivity**
The sensitivity of TCO per degree change in the Niño 3.4 index is another measure of the ozone
response to ENSO determined by the regression analysis.  The spatial distribution of the
sensitivity is shown in Fig. 4.  Over the time period studied here, we find the response to be linear
with respect to the ENSO forcing.  The large region of negative sensitivity in the central Pacific
corresponding to the maximum in explained variance is a result of the increased lofting of ozone-
poor air into the middle and upper troposphere under El Niño conditions.  Thus, higher values of
the Niño 3.4 index correspond to decreases in the TCO.  The opposite sensitivity is found in the
equatorial symmetric lobes over Indonesia and the eastern Indian Ocean where the increased
lofting (decreased TCO) occurs with La Niña (negative Niño 3.4 values).  In the subtropics,
positive sensitivity is located between about 20° and 30° to the north and south of the large
central Pacific minimum.  In addition, relatively strong negative sensitivity exists over South
Africa corresponding to the significant variance explained there.  In the midlatitudes, a negative
albeit weaker response is seen over the United States.  Statistically significant negative responses
are also found over the northern Pacific and Atlantic Oceans.
**3.3   Changes in advection**
The manner by which ENSO impacts the TCO is not well established by previous studies for
regions relatively far removed from the tropical Pacific ENSO oscillations of sea surface
temperatures.  We examine the differences in circulation patterns for strong El Niño and La Niña
conditions to investigate the large-scale impact of the extratropical circulation relative to the
ozone sensitivity.  The streamlines of the difference in the mean winds at 200 hPa for months



with Niño 3.4 index of greater than 0.75 and less than -0.75 are overlaid on the ozone sensitivity
contours in Fig. 4. In the Northern Hemisphere extratropics, anomalous cyclonic circulations
coincide with the regions of negative sensitivity over central Asia, the north Pacific, United
States, and the north Atlantic. The north Pacific and United States circulations agree well with
ENSO-associated upper-troposphere height anomalies observed by Mo and Livezey (1986) and
Trenberth et al. (1998). Similar cyclonic circulations aligned with negative sensitivity in the
Southern Hemisphere are seen over the southern Pacific Ocean and over the southern tip of South
America. Similarly, anomalous anticyclonic flow is associated with positive sensitivity over
much of the midlatitudes.
The meridional and vertical cross-section streamlines of the difference between the mean winds
between 180° W and 120° W for months with Niño 3.4 index greater and less than 0.75 and -0.75
respectively are shown in Fig. 5. The positive and negative sensitivity patterns in this region
shown in Fig. 4 coincide with the anomalous tropospheric downwelling and upwelling. In the
tropics, the anomalous upwelling lofts ozone-poor air into the mid- and upper-troposphere in
agreement with previous studies. Northward of about 40° N, the tropospheric upwelling
coincides with the cyclonic circulation and negative sensitivity shown in in Fig. 4. This is
consistent with increased upwelling induced by cyclonic circulation. Similarly, other anomalous
cyclonic circulations associated with negative sensitivity over North America, the north Atlantic,
and the southern tip of South America also correspond to regions of increased upwelling (not
shown). The positive sensitivity between about 15° N and 30° N corresponds with increased
downwelling and evidence of increased cross-jet transport from the stratosphere into the
troposphere in Fig. 5. Oman et al. (2013) find a similar positive sensitivity in this region and also
in the Southern Hemisphere subtropics in a GEOS-5 CCM simulation. In addition, Lin et al.
(2014) find that increases in springtime ozone following El Niño at the Mauna Loa Observatory
in Hawaii correspond to increased influence by Asian pollution. Here, the relative role of ozone-
rich pollution transport cannot be distinguished from the cross-jet transport since emissions are
not explicitly implemented in the assimilation. The extension of positive sensitivity contours
upstream into the western Pacific to Asia in Fig. 4 is consistent with an influence by Asian
emissions. However, El Niño and La Niña tend to peak in the Northern Hemisphere winter
months when the emissions are least, which would reduce the potential influence.



The qualitative interpretation of the upwelling and downwelling shown in Fig. 5 is supported by
comparison with the dynamical ozone tendency output by the assimilation system. Fig. 6 shows
the differences of the mean dynamical ozone tendencies averaged between 180° W and 120° W
for strong El Niño and La Niña months (the black line). The greatest differences occur in the mid
to upper troposphere, so the net ozone tendencies are shown for the region between the
tropopause and 350 hPa below the tropopause, which provides a constant mass comparison. In
the tropics, the El Niño – La Niña difference in the dynamical tendencies ranges between -0.2 to -
0.55 DU day$^{-1}$, consistent with greater upward transport of ozone-poor air during El Niño than La
Niña. In the lower extratropics, the dynamical tendency differences increase to around 0.2 DU
day$^{-1}$, corresponding with positive ENSO sensitivity in these regions and increased ozone during
El Niño. Negative values of about -0.1 DU day$^{-1}$ exist between 40° and 50° latitude that
correspond with negative sensitivity and upwelling. The small magnitudes at these latitudes are
about 1/6 of the maximum tropical magnitude, which is consistent with the ratio of the
sensitivities in these regions.
The positive sensitivity in the tropics around Indonesia corresponds with increased upwelling
during La Niña conditions rather than with El Niño. This is evident in the downward oriented
streamlines in Fig. 7 showing the circulation differences averaged between 85° E and 120° E for
strong El Niño – La Niña months. In the tropics, the magnitude of the difference is smallest near
the equator, resulting in the northern and southern tropical lobe structure of sensitivity maxima
seen in Fig. 4. The difference is greater in the Southern Hemisphere and the streamlines indicate
more stratosphere to troposphere transport than in the Northern Hemisphere as a possible reason
for the greater sensitivity in the southern lobe located around 15° S.
**3.4 Changes in convection**
In addition to the resolved advective vertical transport and stratosphere to troposphere transport,
TCO can also respond to ENSO through changes in the vertical transport due to convection and
mean depth of the tropospheric column (the tropopause height). This subsection examines the
potential impact from convection using differences in OLR as a proxy. Changes in the
tropopause height are presented in the following subsection.



The differences in the mean OLR for months with Niño 3.4 indices greater and less than 0.75 and
-0.75 over the nine years are shown in Fig. 8. The central Pacific is dominated by decreased OLR
by up to 25%, indicating greater convection under El Niño conditions. The maximum decrease is
displaced to the west of the extrema of explained variance and TCO sensitivity to ENSO (Fig. 3
and 4, respectively). Over the Indonesian region, the OLR is increased by up to 16%, indicating
reduced convection. Here, the maximum OLR changes are offset to the east of the explained
variance and sensitivity extrema.
These spatial offsets suggest that much of the tropical TCO sensitivity to ENSO is realized
through the resolved advective transport. This is supported by the comparison of the analyses
convective and dynamical tendency differences. Fig. 6 compares the El Niño – La Niña
differences in the analysis mid to upper tropospheric convective ozone tendencies (red line) and
dynamical tendencies (black line) between 180° W and 120° W. In the tropics, the convective
tendency differences range from -0.15 to 0.1 DU day$^{-1}$. The dynamical tendency differences are
negative and the magnitudes are more than twice as great as the convective tendency differences.
In the middle latitude north Pacific between 40° N and 50° N, the magnitude of the El Niño – La
Niña convective ozone tendency difference is similar to the dynamical tendency differences (Fig.
6). Thus, the impact on the TCO sensitivity from the resolved transport and convection in this
region are comparable in contrast to the tropics where the resolved transport is dominant.
**3.5    Impact from tropopause height differences**
The sensitivity of the tropopause pressure to the Niño 3.4 index determined by regression
analysis is shown in Fig. 9. The response of the tropopause pressure is generally symmetric
about the equator over the Pacific Ocean. Under El Niño conditions, a slightly greater mean
tropopause pressure (decreased height and shorter tropospheric column) occurs in the extratropics
poleward of the climatological subtropical jet.    Equatorward, decreased tropopause pressures
occur with El Niño, except in the western tropical Pacific where there is a small positive
response. The pattern of tropopause response in the Pacific is similar to the 200 hPa circulation
anomalies in Fig. 4. The offset of the tropical response extrema to the north and south of the
equatorial TCO response (Fig. 4) indicates that very little of the equatorial TCO response is
attributable to changes in the depth of the tropospheric column. The maxima TCO response



around 25° N and 25° S generally coincide with where the tropopause height response is zero.
This also suggests that the positive TCO response here may be impacted by increased
stratosphere to troposphere transport of ozone rich air across the subtropical jet.
Changes in the depth of the tropospheric column associated with ENSO have a greater impact on
the TCO sensitivity in the middle latitudes than in the tropics. Throughout much of the
midlatitudes, positive tropopause pressure sensitivity coincides with negative TCO sensitivity and
vice versa. Particularly noteworthy in the extratropical Northern Hemisphere are the positive
tropopause pressure sensitivity maxima over the northern Pacific, North America, northern
Atlantic, and Asia. The positive and negative tropopause sensitivity over extratropical South
America also aligns closely to the TCO response.
Both the changes in transport (including vertical advection, convection, and cross-tropopause
transport) and the tropopause height can impact the magnitude of TCO. We use regression
analysis of the mean tropospheric mixing ratio on the Niño 3.4 index to make a rough estimate of
the relative influences of transport and tropopause height changes. The mean mixing ratio is
directly sensitive to changes in the transport but not to the tropopause pressure. Note that the
mean mixing ratio also inherently includes any dependence from changes in chemistry that are
associated with ENSO (Sudo and Takahashi, 2001; Stevenson et al., 2005; Doherty et al., 2006).
If the response is assumed linear with respect to changes in transport/chemistry and tropospheric
column depth, the variances explained by the TCO and mean mixing ratio can provide a first
order estimate of the relative roles of these factors. For example, if the TCO explained variance
in a region is 25% and the mixing ratio explained variance is 20%, the tropopause height would
account for an estimated 1/5 of the TCO response.
The spatial pattern of the mean mixing ratio explained variance (not shown) is very similar to the
TCO regression (Fig. 3) in the both the tropics and midlatitudes. Throughout the tropics, the
magnitudes of the variance explained are nearly identical. Thus, changes in transport/chemistry
dominate the TCO response in this region. However, at middle latitudes the explained variance
of mean mixing ratio is frequently less than that of the TCO, so the tropopause height plays a
greater role. For the previously noted Northern Hemisphere negative sensitivity extrema, we
estimate the tropopause height accounts for about a 1/4 of the TCO response to ENSO over the
United States, 1/2 of the response over the North Pacific, and 2/3 of the North Atlantic





sensitivity. The tropopause height is responsible for about 1/5 of the negative sensitivity around
midlatitude South America. Also, only about 1/5 or less of the positive TCO response in the
subtropical Pacific around the climatological subtropical jets is attributable to changes in the
tropopause height.
**3.6    Representativeness of the 9-year assimilation time series**
We use the 22-year (1991-2012) GMI CTM simulation described in section 2.2 to show that the
results from the nine years of assimilation are representative of the longer-term TCO response to
ENSO. The percentage of the simulated TCO variance explained by ENSO during 2005-2012 is
shown in Fig. 10a for comparison with the assimilated ozone results over nearly the same time
period (i.e., Fig. 3). The spatial distribution of the simulated TCO response is very similar. The
maximum variance explained occurs in the central Pacific. The northeast and southeast split
towards Central and South America is evident, but the southern fork is not as prominent. In the
area of Indonesia, the simulated explained variance exhibits the same lobe-like structure
symmetric about the equator. The maximum over the subtropical Pacific and isolated maxima
over the United States and South Africa also agree well with the assimilated ozone results.
Regression analysis of the 22-year time span of the hindcast simulation reveals that much of the
TCO response determined from the nine years of assimilation is consistent with the longer-term
response (Fig. 10b). Use of the longer time series also increases the area in which the explained
variance is statistically different from zero, particularly in the middle latitudes. The shape and
magnitude of the tropical explained variance is similar to the results from the shorter time period.
Two differences are the reduced magnitude extending into the Northern Hemisphere Atlantic and
the slight equatorward shift in the location of the Southern Hemispheric lobe in the Indonesian
region. In the southern subtropical Pacific near 25° S, the maximum in variance explained is
more prominent. Conversely, the maximum in the northern subtropical Pacific is suppressed over
the longer-term. However, there remains an enhancement of greater than 15% explained variance
near 135° W between 15° N and 30° N that is consistent with the shift in the exit region of the
subtropical jet and the associated secondary circulation (Langford, 1999). In the extratropical
northern Pacific, corresponding to the location of negative sensitivity in Fig. 4, the explained
variance is 10%-15% and statistically significant. The signal over the United States and South





Africa persists in the 22-year regression at over 20% explained variance. Over midlatitude
Europe and Asia, the spatial pattern of the explained variance differs between the 22-year and 8-
year regression results. This may be indicative of the variability and trends of emissions being
much more dominant than the ENSO influence in this region.

## 4   Discussion
### 4.1   Tropical response
The tropical tropospheric ozone response to ENSO has been extensively studied in many previous
observational and model investigations. The tropical response in the OMI/MLS ozone analyses
agrees well with these prior investigations and verifies the analyses. However, most studies that
evaluate the spatial distribution of the response do not show a two-lobe structure in the western
Pacific/Indonesian region as seen in the present study (e.g., Ziemke and Chandra, 2003). We
note that this two-lobe structure is also suggested in the ozone sensitivity computed from
Tropospheric Emission Spectrometer (TES) data shown by Oman et al. (2013) in their Fig. 5a.
The symmetric response in this region is likewise well simulated by the GMI CTM driven by
assimilated meteorology (Fig. 10). However, the free-running GEOS-5 Chemistry Climate
Model simulation examined by Oman et al. (2013) produces a single, broad response centered on
the Equator (their Fig. 5b) where the vertical wind differences are consistent with the single,
centered response. This demonstrates that the ozone response is very sensitive to changes in the
advective transport that must be well simulated to reproduce the observed tropospheric response.
### 4.2   Timing of the response
As discussed in section 2, sensitivity tests of possible lags in the ozone response in the regression
analysis did not increase the correlation between the regressed ozone and Niño 3.4 index or
increase the explained variance. In general, the correlation and explained variance remain nearly
constant or decreasing with lag times of one or two months in the middle latitudes. The
correlations generally decrease rapidly with longer lag times. This lack of improved regressions
using longer lag times indicates that there is minimal impact from long-range transport, including
transport in the stratosphere that modulates lower stratospheric ozone concentrations and hence,



the magnitude of large-scale stratosphere to troposphere exchange of ozone. This is consistent
with previous studies that find little relation between ENSO and large-scale stratosphere-
troposphere exchange at midlatitudes (e.g., Hsu and Prather, 2009; Hess et al., 2015). In the
present study, the changes to transport and tropopause height contributing to the TCO response
act over shorter time scales and potentially impact the entire or large portions of the tropospheric
column.
**4.3  Regional aspects of the midlatitude response**
In the middle latitudes, the statistically significant variance explained by ENSO shown in this
study occurs over small-scale regions, so it is not surprising that some previous studies fail to find
an ENSO influence over large-scale regions or in many surface-based observations. For example,
there is no statistically significant explained variance over the midlatitude regions of Canada,
Central Europe, and Japan considered by Hess et al. (2015). These regions also remain
insignificant in the 22-year CTM simulation in the present study.
Conversely, Langford et al. (1998) demonstrate a correlation of ENSO with lidar observations of
ozone near Boulder, Colorado. This coincides with the location of significant explained variance
and negative sensitivity we show in Figs. 3 and 4. However, Langford et al. (1998) show a
positive correlation of mid-tropospheric ozone with the ENSO time series where the ozone signal
lags ENSO by a few months. The lidar ozone anomalies are correlated with the subtropical jet
exit region in the northeastern Pacific (Langford, 1999). He hypothesizes that transverse
circulation across the ENSO-shifted jet exit region brings stratospheric air into subtropical
tropical troposphere where it descends with the secondary circulation and is then transported
northward to the central United States. In the present study, the suggestion of increased localized
stratosphere-to-troposphere transport and subsequent downwelling in the northern subtropical
Pacific is supported by the meridional cross-section of the anomalous wind field (Fig. 5) and the
relatively large TCO response evident in the explained variance and sensitivity (Figs. 3 and 4). It
is possible that episodic events may bring anomalously high ozone air to the central United States
from the subtropics that can impact at least a portion of the tropospheric column. However, we
find that the immediate negative influence by the ENSO-driven vertical transport and tropopause
height changes is dominant when considering the entire tropospheric column.




The negative sensitivity over the United States is consistent with the results of Lin et al. (2015).
They conclude that more frequent springtime stratospheric intrusions following La Niña winters
contribute to increased ozone at the surface and free troposphere in the western United States.
Since the stratospheric intrusions are associated with enhanced stratosphere to troposphere
transport, this can significantly increase the TCO through an influx of ozone-rich air at lower
altitudes.
**4.4   South African region**
We find significant explained variance and sensitivity of TCO around subtropical South Africa.
This is consistent with previous findings. Blalshov et al. (2014) show a correlation of surface
observations of ozone with ENSO. They attribute this association to increased ozone formation
from anthropogenic emissions under warmer and drier conditions occurring with El Niño.
Thompson et al. (2014) remove the ENSO signal from southern Africa region ozonesonde data to
investigate middle tropospheric ozone trends.
Unlike most of the midlatitude TCO response, the processes that drive the TCO response in the
southern Africa region are not clear considering the mechanisms investigated in this study. A
meridional cross-section of the difference in the resolved advective winds averaged between 15°
E and 55° E for strong El Niño and La Niña months (not shown) does not indicate coherent
upwelling consistent with the negative sensitivity found there. Overall, there is weak anomalous
downward transport between about 5 km and 11 km in this region. The differences in OLR (Fig.
8) are also not consistent with unresolved convection as the source of the negative sensitivity.
The tropopause height sensitivity to ENSO in this region (Fig. 9) is positive and similar to the
spatial pattern of TCO sensitivity (Fig. 4) but is weak compared to the relatively strong TCO
response. Therefore, much of the TCO response may be due to ENSO-related changes in the
ozone chemistry that requires further investigation beyond the scope of this study.

**5   Summary**
The assimilation of OMI and MLS data enables this first comprehensive study of the TCO
response along with the ancillary information to interpret and explain the results. We have used





regression analysis of the TCO to provide an observationally-constrained evaluation of the
magnitude and spatial distribution of the ENSO impact on TCO throughout the middle latitudes.
Prior results of the TCO response outside the tropics have been contradictory and limited by the
spatial distribution and sparseness of available data. The present study is able to unify and explain
many aspects of the seemingly disparate findings reported by previous studies.
While the examination of the response in the tropics is included primarily for completeness and
verification of the analyses, two results in this region are novel to this study.  We find that
changes in the large-scale transport dominate the changes in convective transport to produce the
TCO response throughout much of the tropics.  We also show a two-lobe response in the region
around Indonesia that is symmetric about the Equator with maxima near 15° N and 15° S.
The midlatitude ozone response to ENSO is not as strong as in the tropics.  However, the
explained variance is statistically significant over several small regions for the 9-year analysis,
such as over the United States and south of New Zealand.  Other areas have an explained
variance of greater than 10% that the 22-year CTM simulation suggests would be statistically
significant with a longer observation period.  These regions include the northern Pacific and
around midlatitude South America.
The TCO sensitivity to ENSO is relatively small but statistically significant over much of the
midlatitudes.  These regions of negative (positive) sensitivity are coincident with anomalous
cyclonic (anticyclonic) circulation.  The anomalous circulations are associated with upwelling
and downwelling that are consistent with the sign of sensitivity.  In addition to the contribution
by transport, changes in the tropopause height can contribute substantially to the middle latitude
TCO response by altering the depth of the tropospheric column.
This study using analyses of OMI and MLS ozone provides the first explicit spatially resolved
characterization of the ENSO influence and demonstrates coherent patterns and teleconnections
impacting the TCO in the extratropics.  Although relatively weak, the ENSO-driven variability
needs to be considered in investigations of midlatitude tropospheric ozone, particularly on
regional scales.  The spatial variability of the TCO response indicates the ENSO influence is
likely statistically insignificant for hemispheric studies or over other broad areas.  However, the
variance explained by ENSO can be 10% or greater over smaller regions like the United States,





midlatitude South America, and South Africa. Thus, it will be important in attributing the
sources of variability and trends in TCO, such as by human-related activity. These results are
potentially useful for evaluating the spatially dependent model response of TCO to ENSO
forcing. In the extratropics, the ENSO signal is convolved with large extratropical circulation
variability from other sources. Thus, additional factors may need to be considered when
evaluating the midlatitude response in free-running models, particularly in ensemble simulations.

**Acknowledgements**
The authors would like to thank Paul Newman, Jerry Ziemke, Luke Oman, Anne Douglass, and
Susan Strahan for helpful discussions. Funding for this research was provided by NASA's
Modeling, Analysis and Prediction Program and by NASA NNH12ZDA001N-ACMAP. The
assimilated data used in this study are available through the Aura Validation Data Center website:
http://avdc.gsfc.nasa.gov. Simulations and assimilation were done at NASA's Climate
Computing Service under awards from HPC. The Niño 3.4 index used in this study is available
from the NOAA Climate Prediction Center at http://www.cpc.ncep.noaa.gov/data/indices/. The
OLR data is provided by the NOAA/OAR/ESRL PSD, Boulder, Colorado, USA, from their web
site at http://www.esrl.noaa.gov/psd/.





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



**Figure captions**
**Fig. 1.** Time series of the Niño 3.4 index (K) from 1991 through 2013. The time period of ozone
analyses is the black line (2005-2013). The red line indicates the additional years covered by the
GMI simulation. Dashed lines are +0.75 and -0.75 that are considered strong El Niño and La
Niña conditions in this study.
**Fig. 2.** The 2005-2013 annual mean TCO (color contours) from the analyses. Black contours
indicate one standard deviation of the deseasonalized TCO expressed as a percent of the annual
mean TCO. Black contour interval is 0.5%.
**Fig. 3.** The deseasonalized TCO variance explained by ENSO from the linear regression over
2005-2013. Crosshatched areas denote where the confidence level of the explained variance
being different from zero is less than 95%. The increment of the white contours is 5%.
**Fig. 4.** The TCO sensitivity to the Niño 3.4 index from the linear regression over 2005-2013
(color contours). The sensitivity is expressed as the change in the TCO per degree change in the
index (DU K$^{-1}$). Crosshatched regions denote where the sensitivity is not statistically different
from zero at the 95% confidence level. White contours are incremented every 0.3 DU K$^{-1}$. The
streamlines show the difference between the mean winds at 200 hPa for months with strong El
Niño conditions (Niño 3.4 index greater than 0.75) minus months of strong La Niña conditions
(Niño 3.4 index less than -0.75). The thickness of the streamlines is scaled to the magnitude of
the difference. Particularly note the midlatitude regions of negative and positive sensitivity
aligned with anomalous cyclonic and anticyclonic circulations, as discussed in the text.
**Fig. 5.** Streamlines of the difference between the mean vertical and meridional winds for months
with strong El Niño conditions minus months of strong La Niña conditions from 2005-2013. The
means are calculated between 180° W and 120° W. The width of the streamlines is proportional
to the magnitude of the difference. The dashed line indicates the mean tropopause pressure for
strong El Nino months. Solid contours are the zonal mean wind for strong El Niño months.
**Fig. 6.** The dynamical (black) and convective (red) ozone tendency differences between months
of strong El Niño and La Niña conditions from the assimilation system over 2005-2013. The
means are calculated between 180° W and 120° W, matching that of Fig. 5.
**Fig. 7.** As in Fig. 5, but averaged between 85° E and 120° E.





**Fig. 8.** Difference in the outgoing longwave radiation (OLR) for months with strong El Niño
conditions minus months of strong La Niña conditions from 2005-2013. The differences are
expressed as percent of annual mean OLR. Thin white lines are incremented every 2%.
**Fig. 9.** The sensitivity of tropopause pressure to the Niño 3.4 index from linear regression over
2005-2013. The sensitivity is expressed as the change in tropopause pressure per degree change
in the index (hPa K$^{-1}$). Crosshatched regions denote where the sensitivity is not statistically
different from zero at the 95% confidence level. White contours are incremented every 2 hPa K$^{-}$
$^{1}$.
**Fig. 10.** The deseasonalized TCO variance explained by ENSO in the GMI CTM simulation for
years (a) 2005-2012 and (b) 1991-2012. Crosshatched areas denote where the confidence level of
the explained variance being different from zero is less than 95%. The increment of the white
contours is 5%.





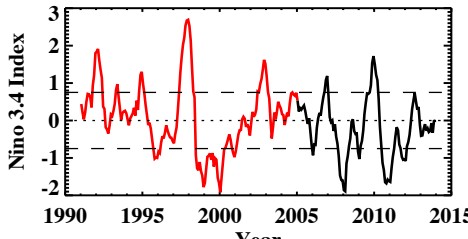

Figure 1. Time series of the Niño 3.4 index (K) from 1991 through 2013. The time period of ozone analyses is the black line (2005-2013). The red line spans the additional years covered by the GMI simulation. Dashed lines are +0.75 and -0.75 that are considered strong El Niño and La Niña conditions in this study.



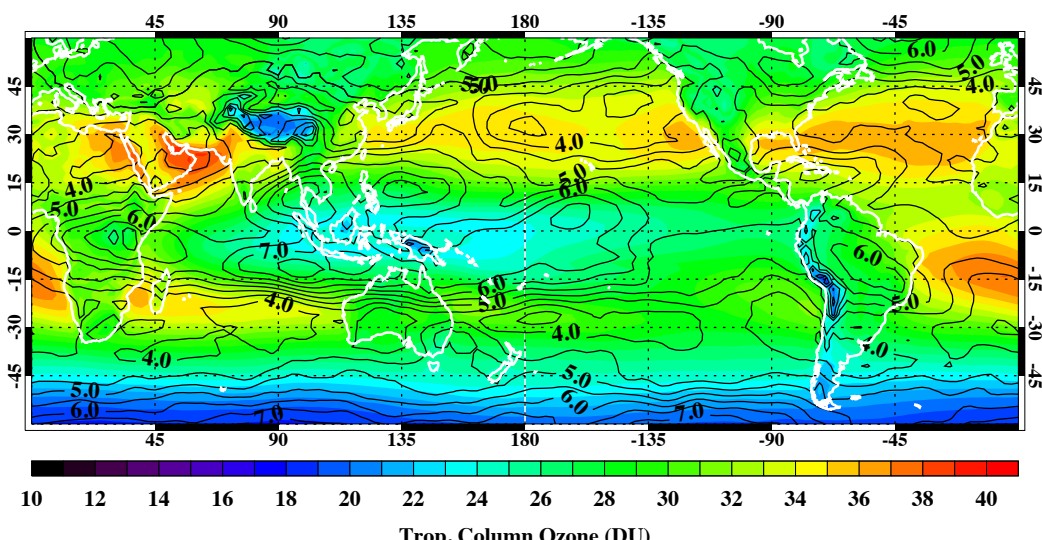

Figure 2. The 2005-2013 annual mean TCO (color contours) from the analyses. Black contours indicate one standard deviation of the deseasonalized TCO expressed as a percent of the annual mean TCO. Black contour interval is 0.5%.





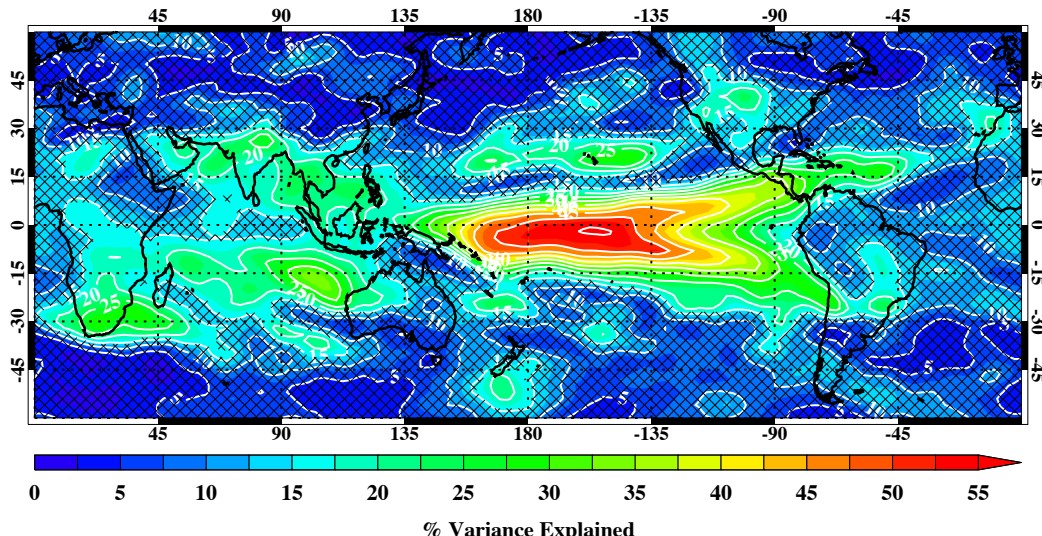

Figure 3. The deseasonalized TCO variance explained by ENSO from the linear regression over 2005-2013. Crosshatched areas denote where the confidence level of the explained variance being different from zero is less than 95%. The increment of the white contours is 5%.





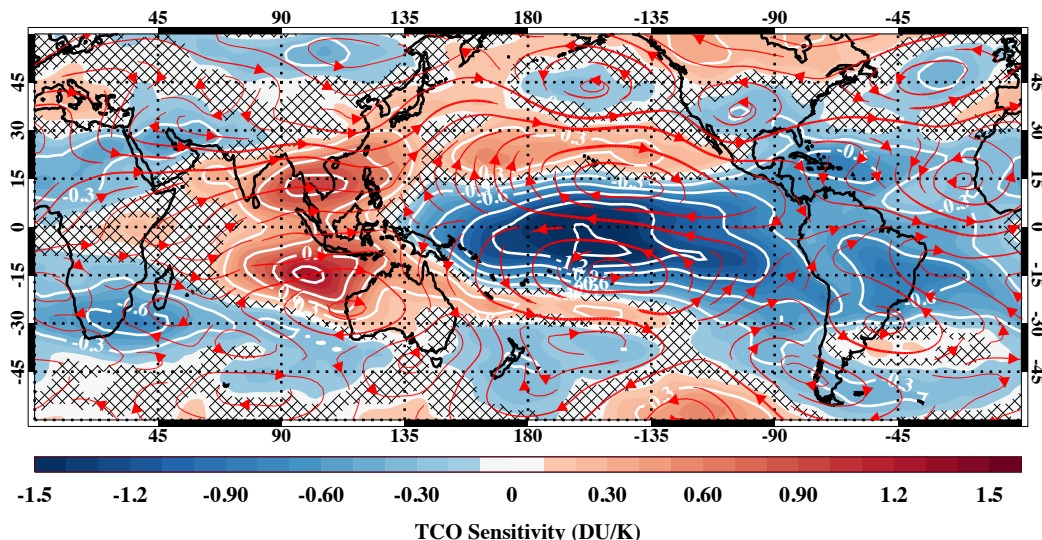

Figure 4. The TCO sensitivity to the Niño 3.4 index from the linear regression over 2005-2013 (color contours). The sensitivity is expressed as the change in the TCO per degree change in the index (DU/K). Crosshatched regions denote where the sensitivity is not statistically different from zero at the 95% confidence level. White contours are incremented every 0.3 DU/K. The streamlines show the difference between the mean winds at 200 hPa for months with strong El Niño conditions (Niño 3.4 index greater than 0.75) minus months of strong La Niña conditions (Niño 3.4 index less than -0.75). Particularly note the midlatitude regions of negative and positive sensitivity aligned with anomalous cyclonic and anticyclonic circulations, as discussed in the text.





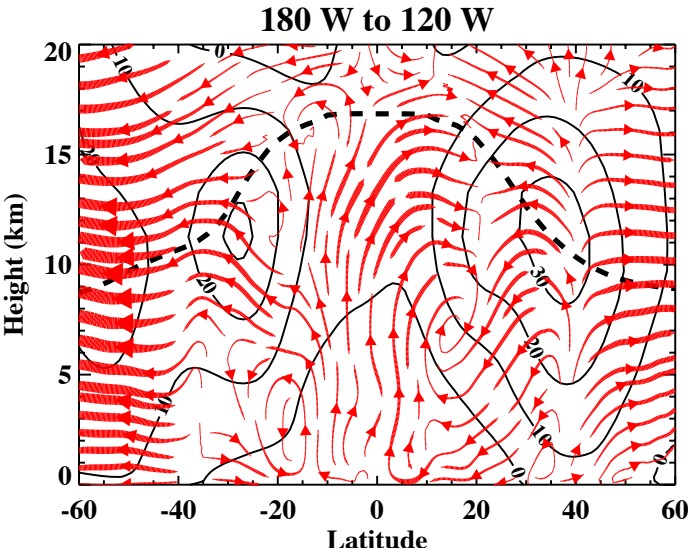

Figure 5. Streamlines of the difference between the mean vertical and meridional winds
for months with strong El Niño conditions minus months of strong La Niña conditions from
2005-2013. The means are calculated between 180° W and 120° W. The width of the
streamlines is proportional to the magnitude of the difference. The dashed line indicates
the mean tropopause pressure for strong El Nino months. Solid contours are the zonal
mean wind for extreme El Niño months.



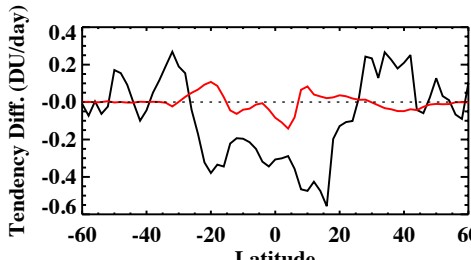

Figure 6. The dynamical (black) and convective (red) ozone tendency differences between months of strong El Niño and La Niña conditions from the assimilation system over 2005-2013. The means are calculated between 180° W and 120° W, matching that of Figure 4.





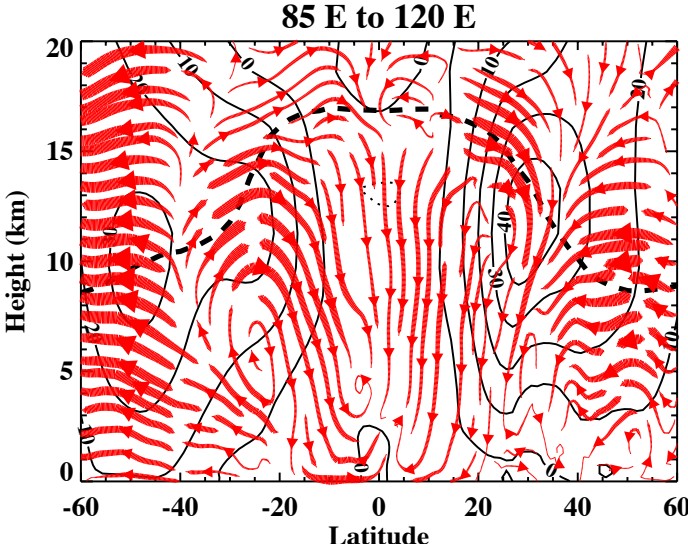

Figure 7.  As in Figure 4, but averaged between 85° E and 120° E.





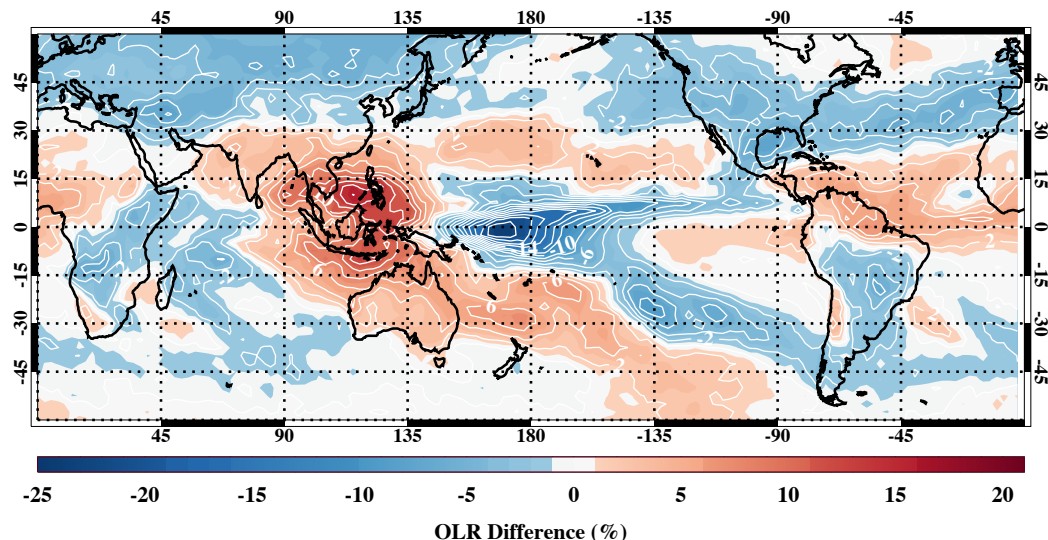

Figure 8. Difference in the outgoing longwave radiation (OLR) for months with strong El Niño conditions minus months of strong La Niña conditions from 2005-2013. The differences are expressed as percent of annual mean OLR. Thin white lines are incremented every 2%.





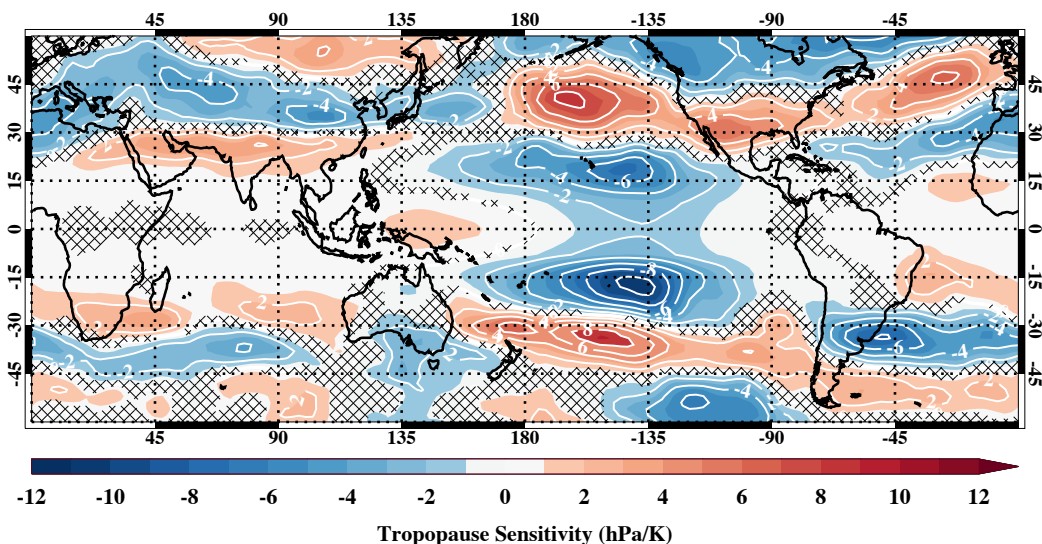

Figure 9. The sensitivity of tropopause pressure to the Niño 3.4 index from linear regression over 2005-2013. The sensitivity is expressed as the change in tropopause pressure per degree change in the index (hPa/K). Crosshatched regions denote where the sensitivity is not statistically different from zero at the 95% confidence level. White contours are incremented every 2 hPa/K.





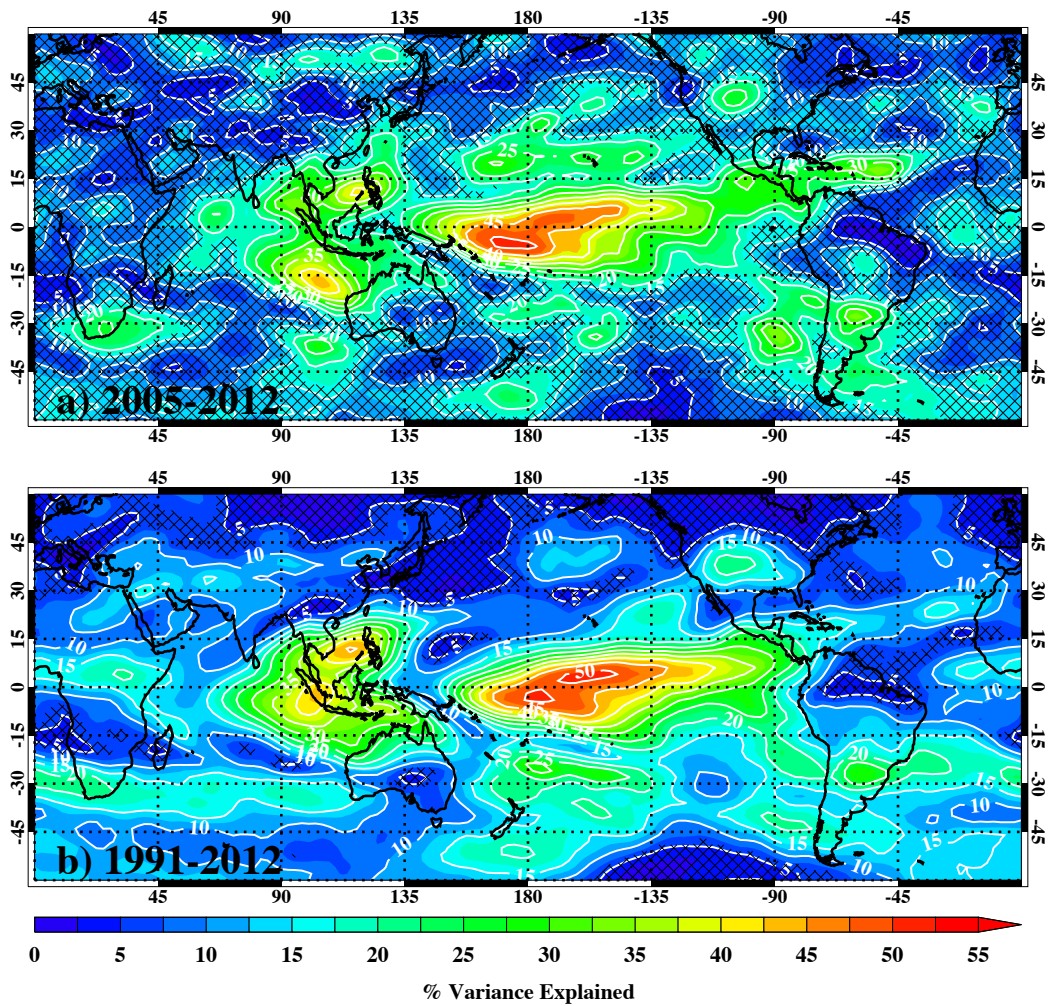

Figure 10. The deseasonalized TCO variance explained by ENSO in the GMI CTM simulation for years (a) 2005-2012 and (b) 1991-2012. Crosshatched areas denote where the confidence level of the explained variance being different from zero is less than 95%. The increment of the white contours is 5%.