# Peer review of "Tropospheric column ozone response to ENSO in GEOS-5 assimilation of OMI and MLS ozone data"

_Atmospheric Chemistry and Physics, 2015_

## Short Comment (SC1) · 29 Jan 2016

"Tropospheric column ozone response to ENSO in GEOS-5 assimilation of OMI and MLS ozone data" by Olsen, Wargan and Pawson is a nice study that investigates the impact of ENSO on ozone in both the tropics and midlatitudes, making use of a 9-year GEOS-5 assimilation and a 22-year CTM simulation.

One point that I would like to dispute, however, is the claim in the abstract of "a newly-identified two-lobed response symmetric about the equator in west Pacific / Indonesia region consistent with large scale vertical transport." Section 4.1 mentions that in Oman et al. (2013) the structure was observed by TES, but not in the model and Lines 492-496, goes on to conclude that this symmetric two-lobed response of ozone in the region is a "novel" finding.

Since they already provide one example of evidence of the two-lobes observed in TES O3, use of the phrase "novel" should be avoided. Two other examples of studies that found these two lobes are Chandra et al. (2009) and Nassar et al. (2009), both of which they have failed to cite anywhere in their manuscript. These studies showed the two-lobe pattern in ozone anomalies by taking the difference of 2006 and 2005 tropospheric ozone. Chandra et al. (2009) used OMI and MLS data along with the GMI model, while Nassar et al. (2009), used TES and GEOS-Chem. In Nassar et al. (2009), we identified that a two-lobe pattern symmetric about the equator, most evident in December anomalies, is primarily of dynamical origin, while fire emissions (via CO oxidation) contributed a single-lobe pattern primarily in October and November (see figure 8). I would suggest updating the manuscript by removing the word "novel" and the phrase "newly-identified" as well as adding a very brief description of the two studies mentioned here.

References

Chandra, S., et al. (2009), Effects of the 2006 El Nino on tropospheric ozone and carbon monoxide: Implications for dynamics and biomass burning, Atmos. Chem. Phys., 9, 4239– 4249.

Nassar, R., J. A. Logan, I. A. Megretskaia, L. T. Murray, L. Zhang, and D. B. A. Jones (2009), Analysis of tropical tropospheric ozone, carbon monoxide, and water vapor during the 2006 El Nino using TES observations and the GEOS-Chem model, J. Geophys. Res., 114, D17304, doi:10.1029/2009JD011760.

---

## Referee Comment (RC1) · Anonymous Referee #3 · 16 Feb 2016

General/Overall Comments

The work focuses on natural phenomena, specifically El Niño Southern Oscillation (ENSO), and its effects on tropospheric ozone with an emphasis on extratropics. The article provides well-rounded background by stressing the importance of separating natural signal from the anthropogenic signal when analyzing tropospheric ozone variability. For the analysis of the tropospheric ozone, the study uses NASA's GEOS-5 data assimilation system (DAS) along with Ozone Monitoring Instrument (OMI) and Microwave Limb Sounder (MLS) on the Earth Observing System Aura satellite. The study also utilizes Global Modeling Initiative (GMT) chemical transport model (CTM) to show that 9 years of ozone assimilation (2005-2013) are consistent with the longer-term tropospheric ozone response. The ENSO is represented by Niño 3.4 index. Outgoing longwave radiation (OLR) data is used as a proxy for convection, which affects tro-

pospheric ozone variability and in turn is influenced by ENSO. Tropospheric column ozone (TCO) is then presented as monthly mean time series and modeled as a function of Niño 3.4 index time series using multiple linear regression with harmonics. The work concludes that large-scale transport seems to dominate the changes in convective transport thereby affecting ozone throughout much tropics. However, effects of ENSO on TCO are much less pronounced in midlatitudes than in tropics.

Overall the article is well-written and the topic of the investigation is clearly established. The article makes an important point that although the effects of ENSO on TCO are generally small in midlatitudes, they are imperative to consider when modeling is performed studying TCO anthropogenic vs. natural variability. I think the article could be improved by adding a few references regarding spatial characterization of the ENSO influence on TCO. For the future work it may be interesting to look at different layers of free tropospheric ozone and to investigate how they respond to ENSO. I approve this article for publication with a few minor suggestions.

Specific Suggestions

On line 41 it states, "This study provides the first explicit spatially resolved characterization. . ." This is a strong statement and probably should be backed up with some kind of reference. The same occurs on lines 509-511.

Lines 56-57 need references.

In lines 66-67, perhaps Thompson et al. 2014 can go to the next paragraph (starting from line 76 and onward) as they actually do not find strong correlation between free tropospheric ozonesonde data and ENSO, while Balashov et al. 2014 do indeed find strong correlation between surface ozone and ENSO.

In line 192, what about a trend in ozone? It may be a good idea to detrend TCO monthly mean time series to see purer ENSO signal in the ozone data.

Perhaps Figures 1 and 6 could be larger?

Lines 251-253 need references.

Technical Comments

In line 420 remove the word "very."

---

## Referee Comment (RC2) · Anonymous Referee #4 · 19 Feb 2016

The authors use ozone observations from OMI and MLS to investigate the ENSO influence on tropospheric column ozone (TCO), including in mid-latitudes. They evaluate the variance explained and sensitivity of TCO to Nino 3.4 index, and find strong response of TCO to the ENSO signal in the tropics which agrees well with the previous studies. They also find that the large-scale transport in the tropics dominate the ozone response compared to the small-scale convective transport. A step forward achieved by this study is the quantification of the ozone response to ENSO induced changes in large-scale circulation and in the mean tropopause height, which contribute significantly to the TCO response in mid-latitudes. This is a very well written paper with some thorough and convincing analyses presented, and is highly recommended to be published in ACP. There are only some minor points listed below.

Specific comments:

Page 4, lines 107-108: "In the midlatitudes, . . . ENSO in some regions" - Is this the finding from your study (then it should be in your conclusions) or from existing studies, in which case these should be cited?

Page 5-6, lines 143-146: The description given here is unclear. You write that "some impact from emissions and other tropospheric chemistry sources and sinks is included in the analyses to the extent that each OMI column retrieval is sensitive to tropospheric altitudes"; can you explain what these impact from emissions and other tropospheric chemistry sources and sinks are? Do you mean the OMI column retrieval is sensitive to tropospheric ozone?

Page 6, lines 155-157: Although the simulations have been described somewhere else, it would be useful to briefly describe the chemical scheme used in these model simulations, and related boundary conditions (i.e. what sources and sinks are included in the model?).

Page 6, line 159: "surface emissions" of what?

Page 9, line 249: I wonder why you didn't mention some significant negative response over the Southern Ocean which are quite obvious.

Page 13, line 362-364, it would be easier to understand if you express these relationships in a formula, or re-phrase the sentence.

Technical corrections:

Page 10, line 271, delete "in" after "shown".

Page 13, line 366, delete "the" before "both".

---

## Referee Comment (RC3) · Anonymous Referee #1 · 15 Mar 2016

This study use GEOS-5 analysis of OMI and MLS ozone observations to examine the magnitude and spatial distribution of the ENSO influence on tropospheric column ozone in the tropics and the mid-latitudes. Overall, the results are a nice contribution to the understanding of the connection between ENSO teleconnection and tropospheric ozone variability, although the time period analyzed in the study is quite short (9 years) in a climate standard. The manuscript is within the scope of ACP. However, there are a number of issues in the current manuscript as outlined in my review below. The referee cannot recommend publication of the paper in ACP unless the authors take serious attempt to address these comments in a revised manuscript.

Major comments:

1. Throughout discussions in the manuscript, particularly in the Introduction section

reviewing previous work on the extratropical trop. ozone response to ENSO (Lines 56 – 85), the discussions will be more clear if you could add information on the data and time period analyzed in each study. It is known that the different time periods or the number of El Nino or La Nina events included in the analysis often gives very different correlation results given the large internal variability of the mid-latitude atmosphere. For example, Langford et al. (1998, 1999) noted positive correlations between mid-tropospheric and lower-stratospheric ozone observed at Fritz Peak, Colorado during 1994–1998 (without La Niña years), reflecting higher than neutral ozone levels during the El Niño events of 1994–1995 (weak) and 1997–1998 (strong). Lin et al. (2015, Nature Communications) finds that their model captures the observed relationship ($r2=0.69$) for this short record, but when the entire 1990–2012 period (including both El Niño and La Niña years) is considered, the model indicates little correlation ($r2=0.18$) between mid-tropospheric and lower-stratospheric ozone over the western US. An extension of the Fritz Peak record to 1999 shows that the mid-tropospheric ozone anomaly in April–May is higher following the La Niña winter of 1998–1999 than in either El Niño or neutral conditions (black circles in Fig. 6c of Lin et al., Nature Communications).

By adding the information on the time period and data used, the readers of the paper could get a sense of how robust the results are.

Throughout the manuscript,the authors tend to contrast their analysis with prior work using shorter records, but not with the recent papers that have examined the mechanisms controlling the extratropical ozone sensitivity to ENSO events more carefully using longer observations and model simulations.

2. In the introduction, you should also discuss the findings of Lin M. et al. (2014, Nature Geoscience) and Neu J. et al. (2014, Nature Geoscience) and data used in their analysis. For instance, you could say:

"Using 40 years of ozone observations at Mauna Loa Observatory and a chemistry-

climate model, Lin et al. (2014) identified a strong link between El Nino events and lower tropospheric ozone enhancements over the subtropical eastern Pacific in winter and spring. Lin et al. (2014) attribute this to the eastward extension and the equatorward shift of the subtropical jet stream during El Nino, which enhances the long-range transport of Asian pollution. Using mid-tropospheric ozone observations from TES during 2005-2010, Neu et al (2014) found ... (http://www.nature.com/ngeo/journal/v7/n5/full/ngeo2138.html)"

3. Lines 175-177 and Figures 5, 7, and 8: According to your classification of ENSO events, there are only two El Nino events but five La Nina events. I speculate that this will affect the statistical power of the composite analysis shown in Figures 5-8. Can these events be really characterized as "strong" ENSO events? The boreal fall/winter of 2008/2009 included in your La Nina composite is not even classified as an ENSO event based on the +/- 0.5 threshold used by CPO (http://www.cpc.ncep.noaa.gov/products/analysis_monitoring/ensostuff/ensoyears.shtml)

4. Lines 230: It is not clear what you mean by "ground-based data". Ground-based data of what? UTLS ozone, mid-tropospheric ozone, lower tropospheric ozone, or surface ozone? The sensitivity of ozone to ENSO events can depend strongly on the vertical altitude as demonstrated previously by Lin et al. (2015) using Trinidad Head ozonesonde data and surface ozone observations over the western U.S., which should be also cited here.

Related to this comment, I also agree with the other reviewer that it would be very nice if you could illustrate and discuss show the sensitivity varies with the altitudes. These new results will be a very nice addition to the TCO sensitivity discussed in the current manuscript.

5. Lines 190-192 and Lines 203-206: It is not clear whether the ozone data is deseasonalized before correlating with the ENSO index. If not, the extent which the sensitivity reported in Figures 3 and 4 is influenced by by correlations on the seasonal time scale?

Please discuss.

6. Lines 251-253: This statement is not true. There are a number of recent studies have extensively examined the mechanisms by which ENSO impacts tropospheric ozone over the extratropical regions, i.e. Lin et al. (2014, 2015) and Neu et al. (2014).

7. Figure 10 and associated discussions in the text: It seems like that there is a substantial difference over the subtropical Northeast Pacific. It is surprising that the variance explained by ENSO over the subtropical Northeast Pacific is very weak in the longer record, but analysis of 40 years of observations at Mauna Loa reveals a strong ENSO signature in free tropospheric ozone over this region (Lin et al., 2014, Nature Geosci). Please discuss. Can you also show a comparison similar to Figure 10 but for the sensitivity shown in Figure 4?

---

## Author Response (AR1)

AUTHORS' RESPONSE TO REVIEWERS

We thank the reviewers and Dr. Nassar for their helpful comments that improved the manuscript. The authors' response to the short comment and three reviewers are listed below in order of submission by the reviewers. The reviewers' comments are italicized while responses are in plain text.

**SC1 (Ray Nassar):**

*"Tropospheric column ozone response to ENSO in GEOS-5 assimilation of OMI and MLS ozone data" by Olsen, Wargan and Pawson is a nice study that investigates the impact of ENSO on ozone in both the tropics and midlatitudes, making use of a 9-year GEOS-5 assimilation and a 22-year CTM simulation.*

*One point that I would like to dispute, however, is the claim in the abstract of "a newly-identified two-lobed response symmetric about the equator in west Pacific/Indonesia region consistent with large scale vertical transport." Section 4.1 mentions that in Oman et al. (2013) the structure was observed by TES, but not in the model and Lines 492- 496, goes on to conclude that this symmetric two-lobed response of ozone in the region is a "novel" finding.*

*Since they already provide one example of evidence of the two-lobes observed in TES O3, use of the phrase "novel" should be avoided. Two other examples of studies that found these two lobes are Chandra et al. (2009) and Nassar et al. (2009), both of which they have failed to cite anywhere in their manuscript. These studies showed the two-lobe pattern in ozone anomalies by taking the difference of 2006 and 2005 tropospheric ozone. Chandra et al. (2009) used OMI and MLS data along with the GMI model, while Nassar et al. (2009), used TES and GEOS-Chem. In Nassar et al. (2009), we identified that a two-lobe pattern symmetric about the equator, most evident in December anomalies, is primarily of dynamical origin, while fire emissions (via CO oxidation) contributed a single-lobe pattern primarily in October and November (see figure 8). I would suggest updating the manuscript by removing the word "novel" and the phrase "newly-identified" as well as adding a very brief description of the two studies mentioned here.*

We thank Dr. Nassar for his short comment related to this manuscript. Although Oman et al. (2013) does not particularly note the two-lobed response in their study, we did overlook the results and discussion made by Dr. Nassar and colleagues in their study. They nicely demonstrate the dynamical origin of the two-lobe response and the chemical origin of the single-lobe response occurring earlier in the year. We greatly appreciate Dr. Nassar for pointing this out and we have modified the manuscript accordingly. The references in the abstract and manuscript to the response as new or novel have been removed. In addition, we have added a comparative discussion of the two-lobe response relative to the Nassar et al. and Chandra et al. studies in our Section 4.1.

**RC1 (Referee #3):**

*…Overall the article is well-written and the topic of the investigation is clearly established. The article makes an important point that although the effects of ENSO on TCO are generally small in midlatitudes, they are imperative to consider when modeling is performed studying TCO anthropogenic vs. natural variability. I think the article could be improved by adding a few references regarding spatial characterization of the ENSO influence on TCO. For the future work it may be interesting to look at different layers of free tropospheric ozone and to investigate how they respond to ENSO. I approve this article for publication with a few minor suggestions.*

We thank the reviewer for their helpful suggestions that have improved the manuscript.

*Specific Suggestions*

*On line 41 it states, "This study provides the first explicit spatially resolved characterization. . ." This is a strong statement and probably should be backed up with some kind of reference. The same occurs on lines 509-511.*

We have added "near-global" to the qualification since it is an important distinction the spatial characterization in our study is not confined to the tropics and not limited to regional analysis in the extratropics.  Other studies that show the spatially resolved response in the tropics and regional impacts at higher latitudes are referenced later in the text.  We have also added the phrase, "To the best of our knowledge,…" at the beginning of this statement.

*Lines 56-57 need references.*

This statement has been moved down in the paragraph so that it directly precedes the citations supporting it.

*In lines 66-67, perhaps Thompson et al. 2014 can go to the next paragraph (starting from line 76 and onward) as they actually do not find strong correlation between free tropospheric ozonesonde data and ENSO, while Balashov et al. 2014 do indeed find strong correlation between surface ozone and ENSO.*

The Thompson et al. (2014) discussion has been moved to the next paragraph as suggested.  We've added the statement that they find the correlation is weak even though they remove the ENSO signal from their ozonesonde time series.

*In line 192, what about a trend in ozone? It may be a good idea to detrend TCO monthly mean time series to see purer ENSO signal in the ozone data.*

We chose not to detrend the TCO time series since the TCO can respond to the trend in the ENSO signal itself.

*Perhaps Figures 1 and 6 could be larger?*

The width of these figures was chosen to be close to one column. Although these figures are generally much smaller than the other figures, we feel that using two columns is not needed for the simple line plots compared to the more detailed contour plots, etc.

*Lines 251-253 need references.*

In response to another reviewer, this sentence has been removed and is not really needed here. The edit does not change the main idea of this subsection.

*Technical Comments*

*In line 420 remove the word "very."*

Removed.

**RC2 (Referee #4):**

*...This is a very well written paper with some thorough and convincing analyses presented, and is highly recommended to be published in ACP. There are only some minor points listed below.*

Thank you. We've improved the manuscript following the comments addressed below.

*Specific comments:*

*Page 4, lines 107-108: "In the midlatitudes, . . . ENSO in some regions" - Is this the finding from your study (then it should be in your conclusions) or from existing studies, in which case these should be cited?*

We find that some readers appreciate having high-level results also presented in the introduction, even though those results may be mentioned in the abstract and conclusion. Thus, readers are reminded and aware of conclusions while going through the details in the results sections.  However, we do see how it may have been confusing whether that statement was referring to our study or previous studies.  We modified by explicitly saying that "we show" these results.

*Page 5-6, lines 143-146: The description given here is unclear. You write that "some impact from emissions and other tropospheric chemistry sources and sinks is included in the analyses to the extent that each OMI column retrieval is sensitive to tropospheric altitudes"; can you explain what these impact from emissions and other tropospheric chemistry sources and sinks are? Do you mean the OMI column retrieval is sensitive to tropospheric ozone?*

We have edited this description to describe that although tropospheric chemistry is not implemented in this version of the assimilation system, increases and decreases to ozone due to chemistry will be included in the analyses through the observations, However, it is limited by the decreasing sensitivity of the OMI retrievals towards the surface.

*Page 6, lines 155-157: Although the simulations have been described somewhere else, it would be useful to briefly describe the chemical scheme used in these model simulations, and related boundary conditions (i.e. what sources and sinks are included in the model?).*

We added a couple of sentences about the simulated chemistry used in the GMI CTM.

*Page 6, line 159: "surface emissions" of what?*

We edited to specify the surface emissions as "anthropogenic and biomass burning" emissions.

*Page 9, line 249: I wonder why you didn't mention some significant negative response over the Southern Ocean which are quite obvious.*

The Southern Ocean response is now included in the revised manuscript.

*Page 13, line 362-364, it would be easier to understand if you express these relationships in a formula, or re-phrase the sentence.*

We edited by stating the resulting value would be 5%, which better illustrates the assumed linear, or additive, relationship.   It is also then easier to see that 5%/25% = 1/5, which is the value previously stated.

*Technical corrections:*
*Page 10, line 271, delete "in" after "shown". Page 13, line 366, delete "the" before "both".*

Fixed.  Thank you.

**RC3 (Referee #1):**

*…Overall, the results are a nice contribution to the understanding of the connection between ENSO teleconnection and tropospheric ozone variability, although the time period analyzed in the study is quite short (9 years) in a climate standard. The manuscript is within the scope of ACP. However, there are a number of issues in the current manuscript as outlined in my review below. The referee cannot recommend publication of the paper in ACP unless the authors take serious attempt to address these comments in a revised manuscript.*

Thank you for your comments.  The additional references and discussions suggested have been added to the manuscript as outlined below for each point.

*Major comments:*

*1. Throughout discussions in the manuscript, particularly in the Introduction section reviewing previous work on the extratropical trop. ozone response to ENSO (Lines 56 – 85), the discussions will be more clear if you could add information on the data and time period analyzed in each study. It is known that the different time periods or the number of El Nino or La Nina events included in the analysis often gives very different correlation results given the large internal variability of the mid-latitude atmosphere. For example, Langford et al. (1998, 1999) noted positive correlations between mid-tropospheric and lower-stratospheric ozone observed at Fritz Peak, Colorado during 1994–1998 (without La Niña years), reflecting higher than neutral ozone levels during the El Niño events of 1994–1995 (weak) and 1997–1998 (strong). Lin et al. (2015, Nature Communications) finds that their model captures the observed relation- ship (r2=0.69) for this short record, but when the entire 1990–2012 period (including both El Niño and La Niña years) is considered, the model indicates little correlation (r2=0.18) between mid-tropospheric and lower-stratospheric ozone over the western US. An extension of the Fritz Peak record to 1999 shows that the mid-tropospheric ozone anomaly in April–May is higher following the La Niña winter of 1998–1999 than in either El Niño or neutral conditions (black circles in Fig. 6c of Lin et al., Nature Communications).*

*By adding the information on the time period and data used, the readers of the paper could get a sense of how robust the results are.*

*Throughout the manuscript, the authors tend to contrast their analysis with prior work using shorter records, but not with the recent papers that have examined the*

*mechanisms controlling the extratropical ozone sensitivity to ENSO events more carefully using longer observations and model simulations.*

We have added information about the time periods and data used in the cited studies. We also added additional discussion and comparisons with previous studies that used longer time series. In particular, we compare with the Lin et al. (2014) in Section 3.6, when discussing the reduced variance explained over the Mauna Loa region with our longer 22-year simulation. In Section 4.3, we added discussion comparing with Lin et al. (2015) relative to the ENSO influence over the U.S. This also now includes the Lin et al. comparison to the Langford results as the Referee discusses above.

*2. In the introduction, you should also discuss the findings of Lin M. et al. (2014, Nature Geoscience) and Neu J. et al. (2014, Nature Geoscience) and data used in their analysis. For instance, you could say:*

*"Using 40 years of ozone observations at Mauna Loa Observatory and a chemistry-climate model, Lin et al. (2014) identified a strong link between El Nino events and lower tropospheric ozone enhancements over the subtropical eastern Pacific in winter and spring. Lin et al. (2014) attribute this to the eastward extension and the equatorward shift of the subtropical jet stream during El Nino, which enhances the long-range transport of Asian pollution. Using mid-tropospheric ozone observations from TES during 2005-2010, Neu et al (2014) found … (http://www.nature.com/ngeo/journal/v7/n5/full/ngeo2138.html)"*

This additional discussion has been added to the Introduction.

*3. Lines 175-177 and Figures 5, 7, and 8: According to your classification of ENSO events, there are only two El Nino events but five La Nina events. I speculate that this will affect the statistical power of the composite analysis shown in Figures 5-8. Can these events be really characterized as "strong" ENSO events? The boreal fall/winter of 2008/2009 included in your La Nina composite is not even classified as an ENSO event based on the +/- 0.5 threshold used by CPO (http://www.cpc.ncep.noaa.gov/products/analysis_monitoring/ensostuff/ensoyears.s html)*

Given the nature of regression, we do not correlate with ENSO "events", but rather the ENSO sea surface temperature (SST) anomaly time series as a whole. In our comparison of the wind and tendency differences, we compare months with magnitudes of SST anomalies greater than 0.75. As stated in the manuscript, this is nearly equal to 1 standard deviation of the time series. We define these months as having "strong" El Niño or La Niña conditions. We do not require the 5 consecutive months of these conditions the CPC uses to color code their chart on the website referenced by the reviewer. Given the value we use as a threshold is about +/- 1 standard deviation of the time series as a whole, we believe that this classification of strong conditions is valid for our comparison of the differences. However, we do see where the misunderstanding originated. We erroneously referred to strong "events" in the original manuscript where we defined our threshold in Section 2.3 (while correctly referring to strong conditions elsewhere). Therefore, we have edited and corrected the discussion in Section 2.3. Thank you for bringing it to our attention!

*4. Lines 230: It is not clear what you mean by "ground-based data". Ground-based data of what? UTLS ozone, mid-tropospheric ozone, lower tropospheric ozone, or surface ozone? The sensitivity of ozone to ENSO events can depend strongly on the vertical altitude as demonstrated previously by Lin et al. (2015) using Trinidad Head ozonesonde data and surface ozone observations over the western U.S., which should be also cited here.*

We replaced with "ground station, FTIR, and ozonesonde data" and added the Lin et al. reference. Other discussions in the manuscript already note these individual studies and the data each uses.

*Related to this comment, I also agree with the other reviewer that it would be very nice if you could illustrate and discuss show the sensitivity varies with the altitudes. These new results will be a very nice addition to the TCO sensitivity discussed in the current manuscript.*

We also agree with the other reviewer that it will be nice future work to look at the altitude variation. Wargan et al. (2015) and Ziemke et al. (2014) have validated the analysis TCO and upper tropospheric column relative to vertically integrated sondes and other data. However, the tropospheric profile information has not yet been sufficiently validated compared to observations. Thus, the suggested work is beyond the scope of this manuscript.

*5. Lines 190-192 and Lines 203-206: It is not clear whether the ozone data is deseasonalized before correlating with the ENSO index. If not, the extent which the sensitivity reported in Figures 3 and 4 is influenced by by correlations on the seasonal time scale? Please discuss.*

Yes, the ozone data is deseasonalized. In the previous manuscript, the sentence right before Lines 190-192 stated the large season variability was removed by subtracting the respective nine-year mean for each month. We have edited this to explicitly say "deseasonalize". In Lines 203-206 of the previous manuscript, we already refer to the "deseasonalized TCO".

*6. Lines 251-253: This statement is not true. There are a number of recent studies have extensively examined the mechanisms by which ENSO impacts tropospheric ozone over the extratropical regions, i.e. Lin et al. (2014, 2015) and Neu et al. (2014).*

We have removed the statement from the revised manuscript without altering the point of the paragraph.

*7. Figure 10 and associated discussions in the text: It seems like that there is a substantial difference over the subtropical Northeast Pacific. It is surprising that the variance explained by ENSO over the subtropical Northeast Pacific is very weak in the longer record, but analysis of 40 years of observations at Mauna Loa reveals a strong ENSO signature in free tropospheric ozone over this region (Lin et al., 2014, Nature Geosci). Please discuss. Can you also show a comparison similar to Figure 10 but for the sensitivity shown in Figure 4?*

The CTM simulated sensitivity over the same time period is shown below. As is evident, the tropical and extratropical pattern is very similar to that in Figure 4. (The greatest difference is the positive sensitivity found over equatorial Africa and outflow due to biomass burning). We find that the sensitivity over Mauna Loa for the entire simulation is similar. However, the variability of the TCO is up to 20% greater in this region over the longer time period, which can account for some of the difference. We have added discussion relative to this in Section 3.6 (Lines 465-471).

[revised manuscript text omitted]

Mark Olsen 4/19/2016 3:41 PM

Mark Olsen 4/8/2016 3:01 PM

Mark Olsen 4/8/2016 3:02 PM

Mark Olsen 4/8/2016 3:07 PM

Mark Olsen 4/8/2016 3:07 PM

Mark Olsen 4/8/2016 3:11 PM

Mark Olsen 4/8/2016 3:05 PM

We use a Global Modeling Initiative (GMI) CTM (Strahan et al., 2007; Duncan et al., 2008)
simulation to determine if the results from the nine years of ozone analyses are representative of
the longer term. Stratospheric and tropospheric chemistry are combined in the GMI CTM with
124 species and over 400 chemical reactions. The tropospheric chemistry mechanism is a
modified version originally from the GEOS-CHEM CTM (Bey et al., 2001). The simulation is
driven using MERRA meteorological fields for 1991-2012 and run at the same resolution as the
assimilation system. Observation-based, monthly-varying anthropogenic and biomass burning
emissions are used through 2010 with repeated 2010 monthly means for the final two years.
Strode et al. (2015) provide more details on this specific simulation, which they refer to as the
"standard hindcast simulation" in their study. Ziemke et al. (2014) show that the TCO from a
similar GMI simulation compares well with sonde observations. In the present study we define,
process, and analyze the CTM TCO fields in the same manner as the assimilation fields.

**2.3 ENSO index and outgoing longwave radiation data**

ENSO is characterized in this study by the monthly mean Niño 3.4 index available from the
NOAA Climate Prediction Center (http://www.cpc.ncep.noaa.gov/data/indices/). The index is
based upon the mean tropical sea surface temperature between $5°$ N – $5°$ S and $170°$ W – $120°$ W.
This time series is normalized using 1981-2010 as the base time period. Fig. 1 shows the index
time series from 1991-2013, which spans the years of the ozone analyses and GMI simulation. In
this study, we define months with "strong" El Niño and La Niña conditions as months with index
values greater than 0.75 and less than -0.75, respectively. The Climate Prediction Center uses
threshold values of 0.5 and -0.5 to characterize El Niño and La Niña, respectively. The value of
±0.75 used here to characterize months of "strong" conditions is about one standard deviation
(0.78) of the time series spanning the assimilation, 2005-2013. La Niña conditions were
dominant during the ozone analyses time period (black line in Fig. 1). Months of strong El Niño
conditions occurred in the boreal fall/winter of 2006/2007 and 2009/2010. Months of strong La
Niña conditions occurred during the boreal fall/winter of 2005/2006, 2007/2008, 2008/2009,
2010/2011, and 2011/2012.

We use outgoing longwave radiation (OLR) data as a proxy for convection to investigate the
contribution from changes in convection associated with ENSO. The monthly, $1° \times 1°$ data is

Mark Olsen 4/8/2016 3:53 PM

Mark Olsen 4/12/2016 12:27 PM

Mark Olsen 4/12/2016 12:28 PM

Mark Olsen 4/12/2016 12:28 PM

Mark Olsen 4/12/2016 12:34 PM

Mark Olsen 4/12/2016 12:34 PM

[revised manuscript text omitted]

Mark Olsen 4/8/2016 11:39 AM

Mark Olsen 4/8/2016 11:56 AM

Mark Olsen 4/8/2016 11:40 AM

Mark Olsen 4/8/2016 11:40 AM

Mark Olsen 4/8/2016 12:04 PM

Mark Olsen 4/8/2016 12:34 PM

Mark Olsen 4/8/2016 12:05 PM

Mark Olsen 4/8/2016 12:34 PM

Mark Olsen 4/11/2016 2:18 PM

Mark Olsen 4/8/2016 12:34 PM

correlations generally decrease rapidly with longer lag times. This lack of improved regressions
using longer lag times indicates that there is minimal impact from long-range transport, including
transport in the stratosphere that modulates lower stratospheric ozone concentrations and hence,
the magnitude of large-scale stratosphere to troposphere exchange of ozone. This is consistent
with previous studies that find little relation between ENSO and large-scale stratosphere-
troposphere exchange at midlatitudes (e.g., Hsu and Prather, 2009; Hess et al., 2015). In the
present study, the changes to transport and tropopause height contributing to the TCO response
act over shorter time scales and potentially impact the entire or large portions of the tropospheric
column.

**524 4.3 Regional aspects of the midlatitude response**

In the middle latitudes, the statistically significant variance explained by ENSO shown in this
study occurs over small-scale regions, so it is not surprising that some previous studies fail to find
an ENSO influence over large-scale regions or in many surface-based observations. For example,
there is no statistically significant explained variance over the midlatitude regions of Canada,
Central Europe, and Japan considered by Hess et al. (2015). These regions also remain
insignificant in the 22-year CTM simulation in the present study.

Conversely, Langford et al. (1998) demonstrate a correlation of ENSO with lidar observations of
ozone near Boulder, Colorado from 1993 to 1998. This coincides with the location of significant
explained variance and negative sensitivity we show in Figs. 3 and 4. However, Langford et al.
(1998) show a positive correlation of mid-tropospheric ozone with the ENSO time series where
the ozone signal lags ENSO by a few months. The lidar ozone anomalies are correlated with the
subtropical jet exit region in the northeastern Pacific (Langford, 1999). He hypothesizes that
transverse circulation across an El Niño-shifted jet exit region brings stratospheric air into
subtropical tropical troposphere where it descends with the secondary circulation and is then
transported northward to the central United States. In the present study, the suggestion of
increased localized stratosphere-to-troposphere transport and subsequent downwelling in the
northern subtropical Pacific is supported by the meridional cross-section of the anomalous wind
field (Fig. 5) and the relatively large TCO response evident in the explained variance and
sensitivity (Figs. 3 and 4). It is possible that episodic events may bring anomalously high ozone

Mark Olsen 4/12/2016 11:23 AM

air to the central United States from the subtropics that can impact at least a portion of the
tropospheric column. However, we find that the immediate negative influence by the ENSO-
driven vertical transport and tropopause height changes is dominant when considering the entire
tropospheric column.

Furthermore, the model evaluation by Lin et al. (2015) reproduces the positive correlation over
the Colorado region for the time period studied by Langford et al. (1998), but the correlation is
not evident when they consider the longer time period from 1990 to 2012. They show that more
frequent springtime stratospheric intrusions following La Niña winters contribute to increased
ozone at the surface and free troposphere in the western United States. Since the stratospheric
intrusions are associated with enhanced stratosphere to troposphere transport, this can
significantly increase the TCO through an influx of ozone-rich air at lower altitudes. The
negative sensitivity over the United States shown in the present study is consistent with these
results of Lin et al. (2015).

**4.4 South African region**

We find significant explained variance and sensitivity of TCO around subtropical South Africa.
This is consistent with the findings of Balashov et al. (2014) who show a correlation of surface
observations of ozone with ENSO. They attribute this association to increased ozone formation
from anthropogenic emissions under warmer and drier conditions occurring with El Niño.

Unlike most of the midlatitude TCO response, the processes that drive the TCO response in the
southern Africa region are not clear considering the mechanisms investigated in this study. A
meridional cross-section of the difference in the resolved advective winds averaged between 15°
E and 55° E for strong El Niño and La Niña months (not shown) does not indicate coherent
upwelling consistent with the negative sensitivity found there. Overall, there is weak anomalous
downward transport between about 5 km and 11 km in this region. The differences in OLR (Fig.
8) are also not consistent with unresolved convection as the source of the negative sensitivity.
The tropopause height sensitivity to ENSO in this region (Fig. 9) is positive and similar to the
spatial pattern of TCO sensitivity (Fig. 4) but is weak compared to the relatively strong TCO
response. Therefore, much of the TCO response may be due to ENSO-related changes in the

Mark Olsen 4/12/2016 11:36 AM

Mark Olsen 4/12/2016 11:38 AM
**Moved down [2]:** The negative sensitivity over the United States is consistent with the results of Lin et al. (2015).

Mark Olsen 4/12/2016 11:37 AM

Mark Olsen 4/12/2016 11:38 AM
**Moved (insertion) [2]**

Mark Olsen 4/12/2016 11:18 AM

Mark Olsen 4/11/2016 11:17 AM

Mark Olsen 4/11/2016 11:17 AM

Mark Olsen 4/11/2016 11:21 AM

Mark Olsen 4/11/2016 3:55 PM

[revised manuscript text omitted]